# Microbiomes of Blood-Feeding Triatomines in the Context of Their Predatory Relatives and the Environment

Hassan Tarabai,[a,b] Anna Maria Floriano,[a] Jan Zima, Jr.,[a] Natalia Filová,[a] Joel J. Brown,[a,c,d] Walter Roachell,[e] Robert L. Smith,[f] Norman L. Beatty,[g,h] Kevin J. Vogel,[i] Eva Nováková[a,j]

[a]University of South Bohemia, Faculty of Science, Ceske Budejovice, Czech Republic

[b]Central European Institute of Technology (CEITEC), University of Veterinary Sciences, Brno, Czech Republic

[c]Biology Centre of the Czech Academy of Sciences, Institute of Entomology, Ceske Budejovice, Czech Republic

[d]Cornell University, Department of Entomology, Ithaca, New York, USA

[e]Public Health Command-Central, Fort Sam Houston, San Antonio, Texas, USA

[f]The University of Arizona, Department of Entomology and Desert Station, Tucson, Arizona, USA

[g]University of Florida College of Medicine, Department of Medicine, Division of Infectious Disease and Global Medicine, and Emerging Pathogens Institute, University of Florida, Gainesville, Florida, USA

[h]Emerging Pathogens Institute, University of Florida, Gainesville, Florida, USA

[i]The University of Georgia, Department of Entomology, Athens, Georgia, USA

[j]Biology Centre of the Czech Academy of Sciences, Institute of Parasitology, Ceske Budejovice, Czech Republic

**ABSTRACT** The importance of gut microbiomes has become generally recognized in vector biology. This study addresses microbiome signatures in North American *Triatoma* species of public health significance (vectors of *Trypanosoma cruzi*) linked to their blood-feeding strategy and the natural habitat. To place the *Triatoma*-associated microbiomes within a complex evolutionary and ecological context, we sampled sympatric *Triatoma* populations, related predatory reduviids, unrelated ticks, and environmental material from vertebrate nests where these arthropods reside. Along with five *Triatoma* species, we have characterized microbiomes of five reduviids (*Stenolemoides arizonensis*, *Ploiaria hirticornis*, *Zelus longipes*, and two *Reduvius* species), a single soft tick species, *Ornithodoros turicata*, and environmental microbiomes from selected sites in Arizona, Texas, Florida, and Georgia. The microbiomes of predatory reduviids lack a shared core microbiota. As in triatomines, microbiome dissimilarities among species correlate with dominance of a single bacterial taxon. These include *Rickettsia*, *Lactobacillus*, "*Candidatus* Midichloria," and *Zymobacter*, which are often accompanied by known symbiotic genera, i.e., *Wolbachia*, "*Candidatus* Lariskella," *Asaia*, *Gilliamella*, and *Burkholderia*. We have further identified a compositional convergence of the analyzed microbiomes in regard to the host phylogenetic distance in both blood-feeding and predatory reduviids. While the microbiomes of the two reduviid species from the Emesinae family reflect their close relationship, the microbiomes of all *Triatoma* species repeatedly form a distinct monophyletic cluster highlighting their phylosymbiosis. Furthermore, based on environmental microbiome profiles and blood meal analysis, we propose three epidemiologically relevant and mutually interrelated bacterial sources for *Triatoma* microbiomes, i.e., host abiotic environment, host skin microbiome, and pathogens circulating in host blood.

**IMPORTANCE** This study places microbiomes of blood-feeding North American *Triatoma* vectors (Reduviidae) into a broader evolutionary and ecological context provided by related predatory assassin bugs (Reduviidae), another unrelated vector species (soft tick *Ornithodoros turicata*), and the environment these arthropods coinhabit. For both vectors, microbiome analyses suggest three interrelated sources of bacteria, i.e., the microbiome of vertebrate nests as their natural habitat, the vertebrate skin microbiome, and the pathobiome circulating in vertebrate blood. Despite an apparent influx of environment-associated bacteria into the arthropod microbiomes, *Triatoma* microbiomes

Address correspondence to Eva Nováková, novake01@prf.jcu.cz.

The authors declare no conflict of interest.

retain their specificity, forming a distinct cluster that significantly differs from both predatory relatives and ecologically comparable ticks. Similarly, within the related predatory Reduviidae, we found the host phylogenetic distance to underlie microbiome similarities.

**KEYWORDS** *Triatoma*, Reduviidae, *Ornithodoros*, microbiome, environment

Insect species diversification has been primarily facilitated by the dietary niche expansion bound to insect physiological adaptation and the establishment of symbiotic interactions with bacteria (1, 2). Microbiomes of insects utilizing resource-limited niches (phloem, xylem, wood, or blood) are known to be dominated by specialized, evolutionarily old bacterial associates with biosynthetic capacities that complement the host metabolism to overcome the dietary limitations. While such symbionts may rise from taxonomically divergent groups even in closely related hosts due to frequent evolutionary replacements (3, 4), these dietary constraints have led to remarkable functional convergence. For instance, within blood-sucking Anoplura, symbionts from distant bacterial genera *Lightella*, *Puchtella*, *Riesia*, *Sodalis*, *Neisseria*, and *Legionella* exhibit parallel genome evolution resulting in retention of genes coding for riboflavin biosynthesis (5, 6). Within a broader frame of blood-feeding Metazoa, functional convergence was found among symbionts of unrelated insect groups and between insects and leeches (7) and was, to some extent, suggested for gut microbiomes of vampire bats and vampire finches (8).

While the majority of the approximately 7,000 Reduviidae species are comprised of hemolymphophagous assassin bugs preying on other invertebrates, the subfamily Triatominae, commonly known as kissing bugs and the vectors of *Trypanosoma cruzi*, converted to hematophagy on birds, mammals, and reptiles (9, 10). In some respects, the symbiosis of Triatominae (Reduviidae) resembles that of hematophagous vertebrates. Unlike tsetse flies, keds, lice, or bedbugs, Triatominae have not established specialized symbiosis during their evolution. They maintain relatively simple species-specific gut microbiomes that undergo remarkable ontogenetic development (11). Though there is not an ancient, universally shared microbiome among kissing bug species, kissing bugs in the genus *Rhodnius* do appear to have frequent associations with bacteria in the genus *Rhodococcus* (12), first identified as a symbiont by Wigglesworth (13). Experimental studies have provided some evidence that the association of *Rhodnius prolixus* and *Rhodococcus rhodnii* is somewhat specific, as even closely related *Rhodococcus* species from other kissing bugs cannot substitute for *R. rhodnii* (14).

Both hemolymphophagy and hematophagy are highly specialized feeding strategies on prey/host body liquids that differ in composition. Invertebrate hemolymph is seemingly easily digestible, and is a rich source of free lipids and free amino acids (15). Vertebrate blood as a food source poses major challenges of heme toxicity and insufficient B vitamin content (16). Based on our knowledge from other hematophagous systems, we hypothesize that the Triatominae dietary shift from hemolymph to blood should be reflected in their microbiome structure and/or function. We further expect to reveal sources of bacterial influx associated with the *Triatoma* natural habitat, i.e., host vertebrates and their nesting sites. In this study, we investigate microbiome signatures of five *Triatoma* species against the background of five species of hemolymphophagous assassin bugs. Another vector species, a soft tick, *Ornithodoros turicata*, was included in the study to provide a phylogenetically independent contrast to microbiomes of hematophagous species sharing the same habitat. The analyzed arthropods originate from sympatric populations collected from nests of small vertebrates, primarily the white-throated wood rat, *Neotoma albigula*, in the southern United States. Within this unique sample set, we track the environmental sources of bacteria and address possible taxonomic convergence between microbiomes of sympatric populations of kissing and assassin bugs, and between the two hematophagous vectors, kissing bugs and ticks.

## RESULTS AND DISCUSSION

**Taxonomic determination and distribution of sampled arthropods.** In total, we have molecularly identified five species of *Triatoma* spp. (*T. gerstaeckeri*, n = 7; *T. lecticularia*, n = 12; *T. protracta*, n = 39; *T. rubida*, n = 46; *T. sanguisuga*, n = 6) and four species

of assassin bugs (*Stenolemoides arizonensis*, n = 3; *Ploiaria hirticornis*, n = 10; *Zelus longipes*, n = 7; *Reduvius sonoraensis*, n = 2). For 15 assassin individuals from Las Cienegas National Conservation Area (LCNCA) (AZ), 16S rRNA Sanger sequences contained signals for both the assassin and its prey, and we thus relied on morphological determination to the genus *Reduvius* (see Fig. S1 in the supplemental material). Most of the analyzed *Neotoma* nests housed *Triatoma* sp. (14/16) and *O. turicata* (12/16). While 10 nests allowed for comparisons between microbiomes of blood-feeding vectors, i.e., ticks and triatomines, two nests served for comparisons of predatory and blood-feeding Reduviidae. Another two nests contained some species from all three sample types, i.e., ticks, assassins, and triatomines. In addition, *Zelus longipes* (Harpactorinae) and *Ploiaria hirticornis* (Emesinae: Leistarchini) were found in two locations in Georgia not associated with *Neotoma* nests (Table S1, data sheet S1).

**Quality of rRNA amplicon data.** The MiSeq runs resulted in 5,998,041 sequence pairs with successful merging and trimming of 3,278,636 pairs (54.66%) sharing a mean merged length of 404 bp and an average number of 7,514 reads per sample. The profiles of negative controls were utilized to identify contaminant operational taxonomic units (OTUs) (see Materials and Methods and Table S1, data sheet S4). The sequencing of the positive controls showed a bias against *Rhodobacter* and in favor of *Staphylococcus* species, but the overall frequencies of the sequenced taxa are consistent with their abundance in the input material and comparable across the control replicates (Fig. S2). OTUs (n = 4,777) were clustered from the entire data set, i.e., amplicons retrieved for 255 arthropods, related environmental material, and controls (Table S1, data sheet S5). For the 12S rRNA gene, used for determination of primary and secondary blood meal source (Table S1, data sheet S7), a total of 1,287,068 reads were retrieved within the arthropod data set with an average of 5,297 reads per sample.

**Microbiomes of North American *Triatoma* kissing bugs.** The profiles of *Triatoma* microbiomes analyzed here concur with the characteristics previously published by our team (11). These include significant differences in taxonomic composition and diversity among different *Triatoma* species (Table S1, data sheets S8 to S10) which undergo an ontogenetic shift from a diverse microbiome to a very simple community dominated by a single bacterial genus (Fig. S3). Venn analyses confirmed *Actinobacteria* as the dominant phylum in *Triatoma* microbiomes with the genus *Dietzia* found in over 50% individuals of phylogenetically closely related *T. rubida*, *T. lecticularia*, and *T. protracta*. The same fraction of *T. gerstaeckeri* and *T. sanguisuga* individuals shared two actinobacterial genera, i.e., *Pseudonocardia* and *Nocardioides* (Table S1, data sheet S11).

Regardless of the phylogenetic relationships, *T. gerstaeckeri*, *T. lecticularia*, *T. rubida*, and *T. sanguisuga* shared two proteobacterial taxa, an *Acinetobacter* (OTU_17) and *Serratia* sp. (OTU_14, Table S1, data sheet S11). While we cannot currently assess their role in *Triatoma* microbiomes, it is worth noting that both taxa are commonly found in insects (17, 18) and in some cases provide profound advantages for their host. For instance, some *Acinetobacter* strains are associated with insect resistance to the insecticide cypermethrin (19), a pyrethroid used for control of insect pests (20–22). *Serratia marcescens* isolated from the Triatominae microbiome has negative effects on the survival and replication of *Trypanosoma cruzi* in the *Rhodnius prolixus* gut (23, 24). Notably, a number of other bacterial taxa are almost universally present in *Triatoma* species across the majority of individuals (≥80%), for instance, *Staphylococcus*, *Massilia*, *Bacillus*, *Planococcus*, and *Streptomyces* (Fig. 1). Most were previously reported in microbiomes of some Triatominae species (11, 25, 26). We elucidate the putative origins of these omnipresent taxa below, providing the evolutionary and ecological frame of this study.

**Microbiomes of related predatory assassin bugs.** Similar to *Triatoma*, host species was a significant predictor of assassin bug microbiome alpha (Table S1, data sheet S12) and beta (Fig. 2 and Table S1, data sheets S13 and S14) diversity, explaining over 60% of variation. Clear dissimilarities correlate with dominance of a single bacterial taxon in each assassin bug species, i.e., *Rickettsia* in *Ploiaria hirticornis*, *Lactobacillus* in *Reduvius* sp., "*Candidatus* Midichloria" in *Stenolemoides arizonensis*, and *Zymobacter* in *Zelus longipes* (Fig. 2A). Species specificity and dominance by a single taxon resemble

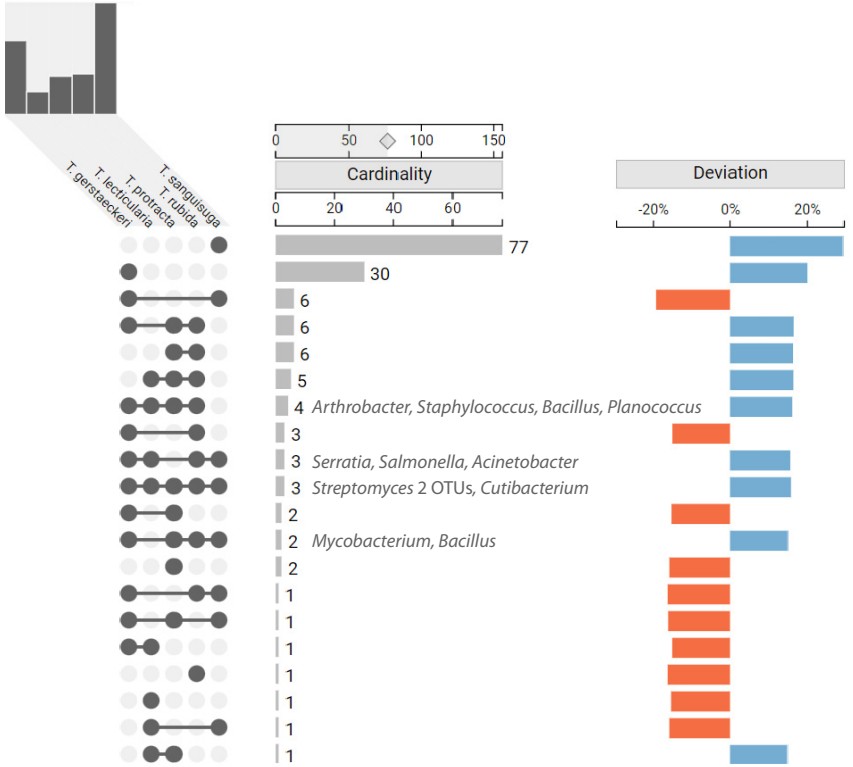

**FIG 1** UpSet graph depicting microbiome intersections among different *Triatoma* species. The top bar chart stands for the cumulative number of OTUs identified for each *Triatoma* species. The colored plot indicates how much each intersection deviates from the expected size if species membership within the analyzed data set is random. The bacterial genera shared, in 0.5 fraction, by four or more species are listed.

the microbiome characteristics of related hematophagous kissing bugs (11) but pose a sharp contrast to the recently published microbiome patterns in Old World assassins (27). Based on the data from six Harpactorini species, Li and colleagues suggested *Enterococcus* bacteria to be a conserved microbiome component in most assassin bugs (27). In our data, *Enterococcus* is, however, found only in some individuals of *Reduvius* sp. and *Stenolemoides arizonensis*, both distantly related to the Harpactorini group of reduviids. For the only Harpactorini species in our data set, *Zelus longipes*, *Enterococcus* is absent across all individuals. The same applies for analyzed *Ploiaria hirticornis* individuals from the Emesinae group. A surprisingly high number of assassin bug-associated bacteria may be of further interest as they belong to known insect symbionts. These are unevenly distributed and include *Rickettsia*, "*Candidatus* Midichloria," *Wolbachia*, "*Candidatus* Lariskella," *Asaia*, *Gilliamella*, *Burkholderia*, and the bacteria from the *Morganellaceae* family (designated "endosymbiont8" [Fig. 2A]) with the closest BLASTn hit being to *Arsenophonus* symbionts known for a broad range of associations with invertebrates (28).

However, the primary goal of this study was not a thorough analysis of microbiome components in predatory Reduviidae. We only provide a habitat-specific frame for investigation of microbiome signatures underlain by the evolutionarily significant dietary switch of Triatominae predecessors. While our data do not suggest the existence of an assassin core microbiome (Venn for 0.5 fraction; Fig. 2B and Table S1, data sheet S15), under less stringent conditions they suggest a possible compositional convergence among *Triatoma* and *Reduvius* species (Table S1, data sheet S16) sharing *Salmonella* and *Serratia*.

**Microbiome of the soft tick *Ornithodoros turicata*.** The microbiome profiles retrieved in this study resemble those described for *O. turicata* from Bolson tortoises in

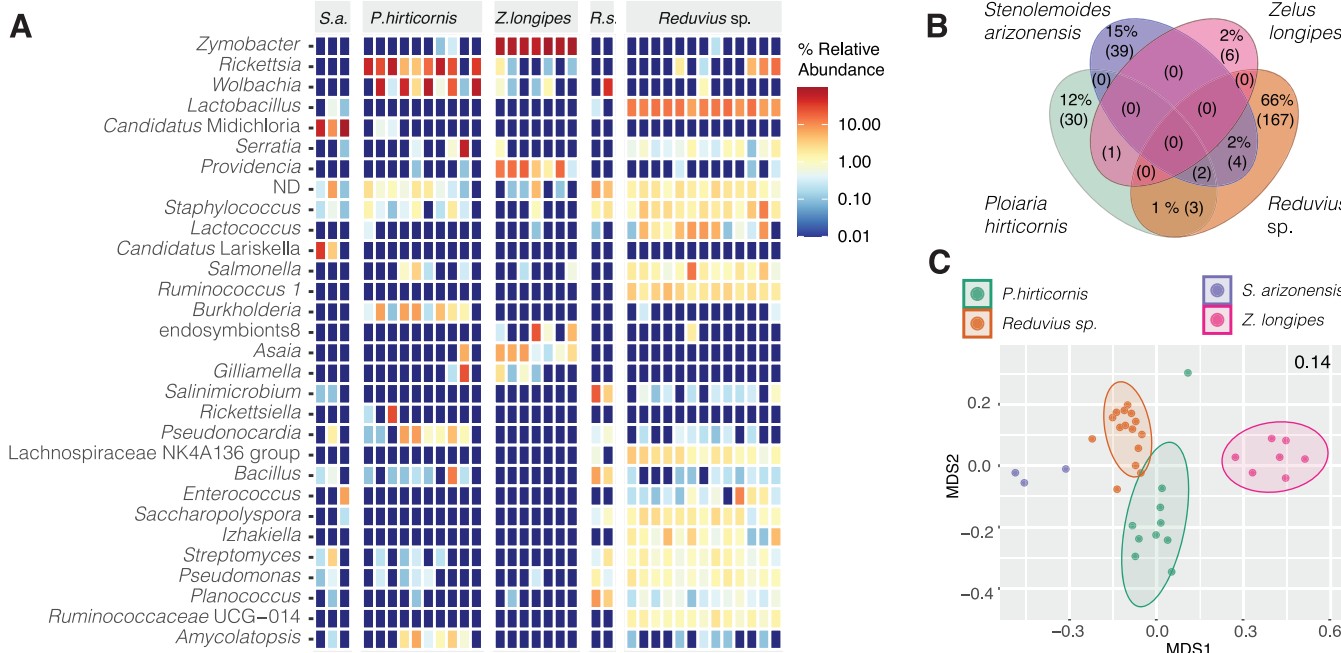

**FIG 2** Microbiome profiles of hemolymphophagous assassin bugs. (A) Heatmap showing the distribution of the 30 most abundant bacterial taxa identified across the assassin species. (B) Shared proportion of the microbiome components among the analyzed assassins. The numbers in the parentheses stand for the absolute number of OTUs. The shared proportion of the microbiome that accounts for less than 1% is depicted only as the absolute count (C). NMDS clustering based on Bray-Curtis distances. The number in the right upper corner indicates the stress value. Two *Reduvius* species were merged under a single group, *Reduvius* sp.

northern Mexico (29). However, *Midichloria*-related symbionts were not the most abundant taxa in our data set. Instead, several individuals harbored two *Candidatus* taxa from the *Midichloriaceae* family, "*Candidatus* Jidaibacter" and "*Candidatus* Lariskella" (Fig. 3A), and microbiomes were dominated by known symbiotic genera of ticks, i.e., *Coxiella* and *Rickettsia* (Fig. 3). While *Coxiella* symbionts have been previously described from *O. turicata* and other *Ornithodoros* species (29, 30), bacteria of the genus *Rickettsia* were identified only as potential symbionts of another soft tick species, *Carios vespertilionis* (30). The distributions of these two genera across our data set differed substantially. The genus *Coxiella* was omnipresent across nests and localities, detected in 96% of analyzed individuals. The genus *Rickettsia* was associated only with 26 individuals sampled in Desert Station, AZ (32% of all the samples). *Rickettsia* presence/absence further reflects the origin of the analyzed ticks from individual *Neotoma* nests and suggests the bacteria may rather represent vertebrate pathogens acquired through feeding. A similar pattern was observed for *Borrelia* pathogens vectored by ticks. *Borrelia* nest-specific distribution was further corroborated by the presence of the pathogen (detectable at a median relative abundance of 0.27%) in *Triatoma* individuals cohabiting the same nests as the infected ticks. Compared among the three sampled localities, microbiomes of Lackland Air Force Base (AFB)-collected ticks displayed significantly higher measures for alpha diversity (Fig. 3C and Table S1, data sheet S17). In the ordination-based analysis, the microbiomes clustered according to their geographic origin (principal-coordinate analysis [PCoA] capturing 16% of the variation) (Fig. 3B and Table S1, data sheet S18, provide statistical support).

**Phylogenetic inference of possibly symbiotic taxa shared by different hosts.** We investigated the phylogenetic origin of possibly symbiotic OTUs from the families "*Candidatus* Midichloriaceae," *Morganellaceae*, *Rickettsiaceae*, and *Acetobacteraceae* detected across different hosts. For the *Rickettsiaceae* OTUs, present in all *P. hirticornis* and *Reduvius* samples, five *Zelus* samples, and 37 *O. turicata* samples, our amplicon data could not provide sufficient phylogenetic resolution (Fig. S5). A similarly uncertain result was obtained for *Acetobacteraceae* OTUs taxonomically assigned to the genus *Asaia*. While OTU_39 falls among symbionts of mosquitos and *Asaia* species isolated

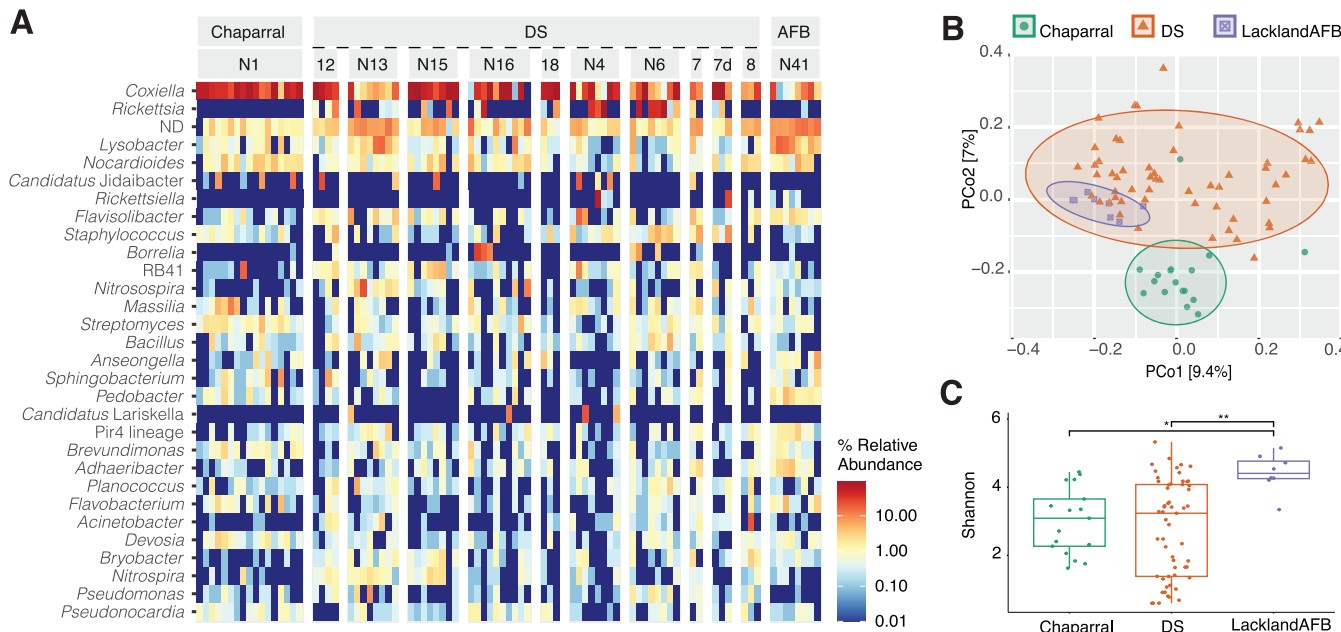

**FIG 3** *Ornithodoros turicata* microbiome profile. (A) Heatmap for the 30 most abundant genera. ND stands for the sum of the unclassified taxa at the genus level. (B) Principal-coordinate analysis based on Bray-Curtis distances. Colored shapes stand for different localities: DS, Desert Station, Tucson, AZ; Chaparral, Chaparral Wildlife Management Area, TX; LacklandAFB, Lackland Air Force Base, San Antonio, TX. (C) Bray-Curtis microbiome dissimilarities calculated among the three localities. Asterisks indicate significant differences at 95%* and 99%** confidence interval found between the diversity measures (Dunn's Kruskal-Wallis multiple comparisons, Table S1, data sheet S17).

from flowers (31, 32) the relationship of two other sequences to the genus remains highly questionable (Fig. S6).

Three *Midichloriaceae* OTUs, found in higher abundances in 14.2% ($n = 5/35$) of the assassin bugs and 20.7% ($n = 17/82$) of ticks, correspond to three distinct lineages (Fig. 4), two of which encompass endosymbionts of arthropods. OTU_6, present in assassin bugs, falls within the *Midichloria* genus, a well-known group of tick endosymbionts (33). OTU_26, found in both assassin bugs and ticks, clusters with the *Lariskella* group, which includes symbionts of ticks and insects (34). Interestingly, OTU_16, present in ticks, clusters with two *Fokinia* species, which reside in aquatic environments as symbionts of *Paramecium* (35, 36). However, this node is not highly supported, including two highly diverging *Midichloriaceae* (35, 36), and might be a product of long branch attraction. For the *Morganellaceae* OTU, we prepared a data set of 154 16S rRNA gene sequences, as in reference 28, including a single outgroup. The inferred phylogeny (Fig. S7) is well supported and clearly shows OTU_67 clustering with *Morganellaceae* symbionts of aphids. Among *Zelus longipes* individuals, the OTU distribution does not, however, suggest that the bacteria play a stable symbiotic role (being detected in 50% of the samples) and may point out its origin in the prey that often includes aphids.

**Blood meal signature among *Triatoma* and tick microbiomes.** Primary blood meal was identified in all five sampled *Triatoma* species (93 individuals, 84%) and five individuals (6%) of *Ornithodoros turicata* (Table S1, data sheet S7). The striking difference in the fractions with determined vertebrate hosts is undoubtedly related to the distinct physiology of triatomines and soft ticks. While the two vectors feed on similar hosts, including mammals, reptiles, and birds (10, 37), soft ticks are known to endure extremely long starvation periods (38) that may hamper the molecular detection of ingested blood meal (39). For *T. protracta* (38/39), *T. rubida* (34/46), and *T. lecticularia* (10/12), packrats (*Neotoma* sp.) were indeed the most common blood meal (82/93, 88%). For six triatomines in our data set (6%), i.e., *T. gerstaeckeri* (4/7) and *T. sanguisuga* (2/6), armadillos, *Dasypus* sp., served as a blood meal source. Five *Triatoma* individuals fed on species of brown-toothed shrews, *Episoriculus* sp. (3), deer mice (*Peromyscus* sp.), and ground squirrels (*Callospermophilus* sp.). For 35 *Triatoma* individuals and a

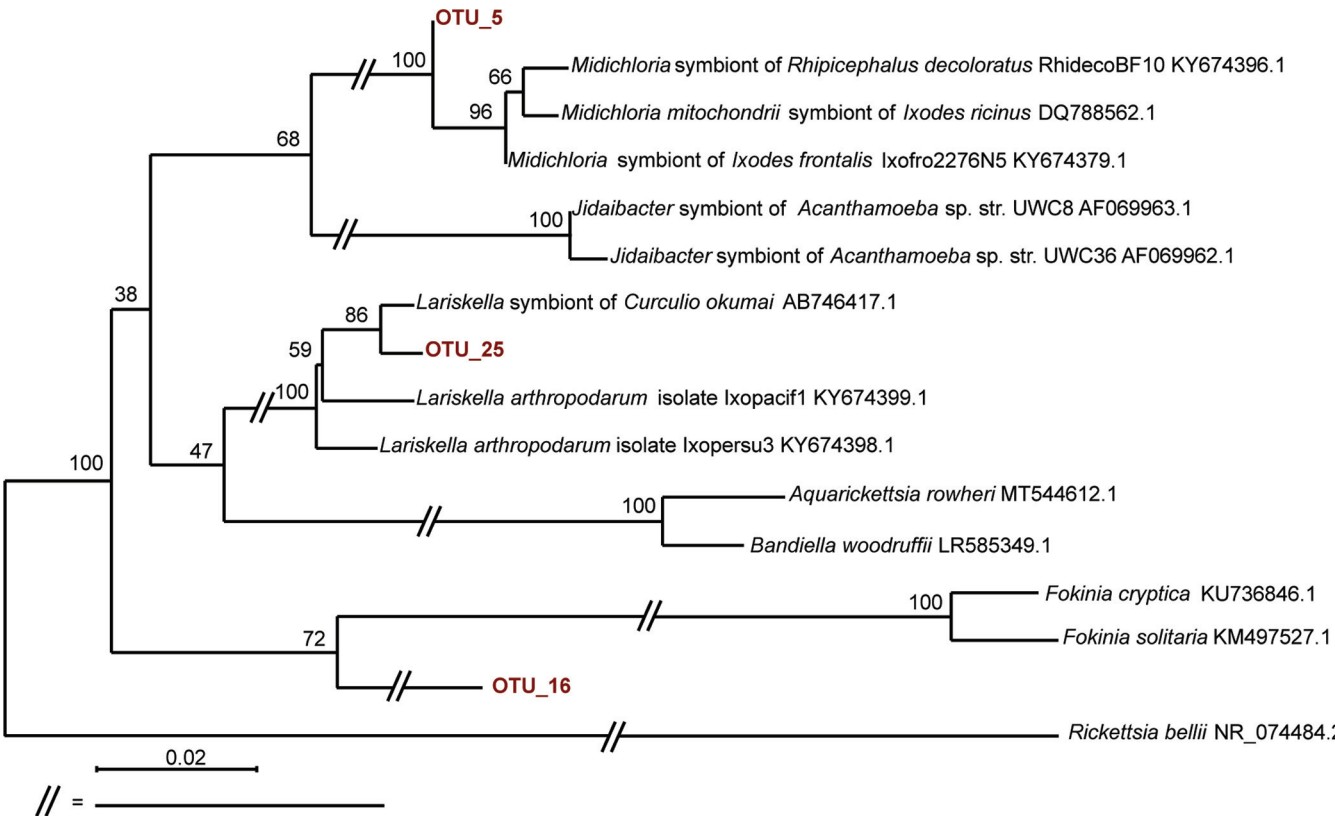

**FIG 4** Phylogenetic inference on 16S rRNA gene sequences of *Midichloriaceae* OTUs associated with assassin bugs and ticks. Numbers at the nodes are bootstrap values.

single tick, we were able to determine secondary blood meal sources (see Materials and Methods). These included the genera *Felis* ($n = 32$) and *Dasypus* ($n = 3$) and individual records on *Callospermophilus* and *Neotoma* (Table S1, data sheet S7). *Neotoma* predominance, along with the absence of human and domestic animals, may well reflect the sampling strategy, i.e., an active search for triatomines in sylvatic packrat nests. The range of other, less frequent hosts listed above is in line with earlier studies on North American *Triatoma* species (40, 41).

The uneven distribution among identified blood sources did not allow us to further evaluate whether different vertebrate hosts may determine the vector microbiome profile, as previously discussed for triatomines (42, 43) and other vectors, e.g., yellow fever mosquitos (44), tsetse flies (44), and western black-legged ticks (45). We therefore searched for common microbiome signatures among those individuals that fed on *Neotoma* (Table S1, data sheet S19). At least 50% of *Neotoma*-fed *Triatoma* and tick individuals harbored *Cutibacterium* (OTU_41), *Streptomyces* (OTU_102 and OTU_295), *Brachybacterium* (OTU_650), *Arthrobacter* (OTU_214), *Microbacterium* (OTU_254), *Pseudonocardia* (OTU_513), *Massilia* (OTU_43), *Pseudomonas* (OTU_59), and *Acinetobacter* (OTU_17), pointing out their potential link to the vertebrate host. Three of the OTUs belong to antimicrobial-producing bacteria (i.e., *Pseudonocardiaceae* and *Streptomycetaceae*) previously identified within the Key Largo woodrat (*Neotoma floridana smalli*) body microbiome and the microbiome of their nests in Florida (46, 47). In addition, *Cutibacterium*, a common skin commensal from the *Propionibacteriaceae* family (48), has been repeatedly recorded in the body swabs from small rodents (unpublished laboratory data). Notably, all six *Triatoma* individuals that fed on *Dasypus* harbored *Alteribacillus* (OTU_118), *Pseudonocardia* (OTU_177), and *Mycobacterium* (OTU_2341). The *Mycobacterium* OTU identified here represents a good candidate for a blood meal-related microbiome signature since *Mycobacterium leprae*, the causative agent of leprosy in humans, is commonly found among nine-banded armadillos (*Dasypus novemcinctus*) in Texas and Florida (49).

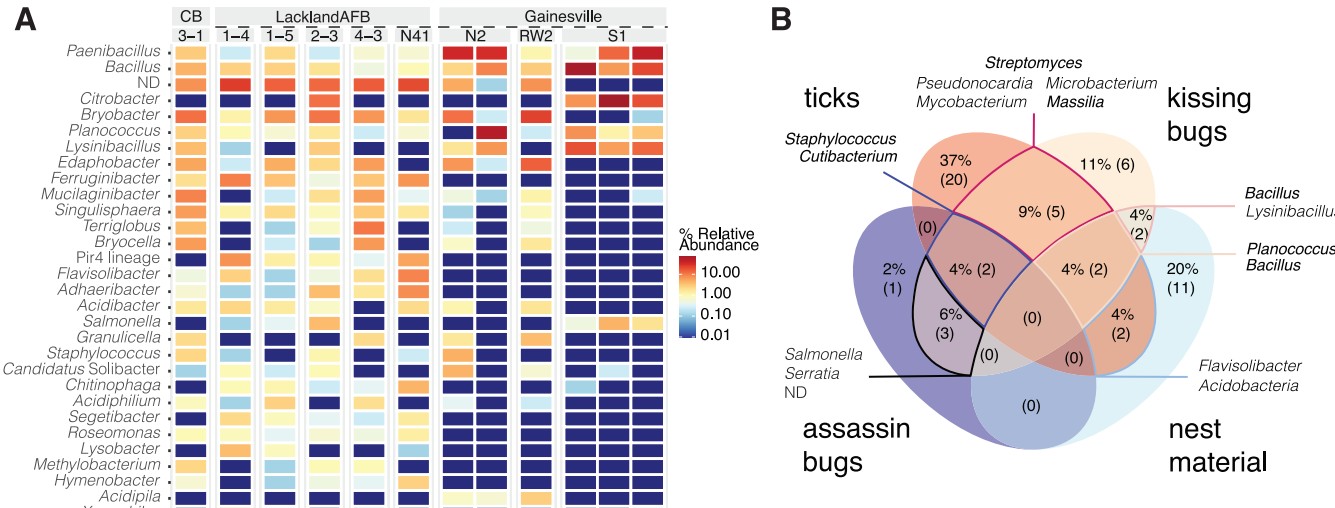

**FIG 5** Nest microbiome characteristics. (A) Heatmap for the 30 most abundant genera across the sampled nests. ND stands for the sum of the unclassified taxa at the genus level. (B) Venn analysis at 0.5 fraction among all sample types, i.e., ticks, kissing and assassin bugs, and nest material. The numbers in the parentheses stand for the absolute number of OTUs. The taxa in boldface were repeatedly recognized as shared microbiome members under different fractions (Table S1, data sheets S22 and S10) and thus considered of nest environmental origin. CB stands for Camp Bullis sampling site (data sheet S1).

**Environmental microbiome.** The evaluation of nest material revealed a significant amount of variation between microbiome diversities of nests sampled in Gainesville, FL, and Lackland Air Force Base (AFB), TX (Adonis2; $R^2 = 0.274$, $P = 0.008$) (Fig. S4 and Table S1, data sheets S20 and S21). The location also determined significant differences found for all alpha diversity measures (Dunn's Kruskal-Wallis multiple comparisons, Table S1, data sheet S22, and Fig. S4). While the nest microbiome from Florida tends to be dominated by a few taxa, especially *Bacilli* genera *Paenibacillus*, *Bacillus*, and *Lysinibacillus*, Texas locations display more diverse and equally structured environmental microbiomes (Fig. 5A) composed mainly of *Acidobacteriia* (*Bryobacter* and *Edaphobacter*), *Chitinophaga* (*Ferruginibacter* and *Flavisolibacter*), and *Planctomycetia* (*Singulisphaera* and *Pirellula*-related Pyr4 lineage). Few taxa are, in different relative abundances, universally present across the sampled nests, i.e., *Paenibacillus*, *Bacillus*, and *Planococcus* (Fig. 5A). As the universal components of nest microbiomes, *Bacillus* (OTU_20) and *Planococcus* (OTU_50) are, at different Venn fractions, also observed in samples of *Triatoma*, ticks, and assassin bugs (Fig. 5B and Table S1, data sheets S23 and S11), supporting a role of the environment in shaping the *Triatoma* microbiomes as suggested previously (11). For triatomines and ticks occupying the same nest, *Massilia* (OTU_43) and several *Streptomyces* OTUs (Fig. 5B) were found as shared microbiome components, corroborating the findings on *Neotoma*-fed vectors above. Two OTUs, *Staphylococcus* (OTU_3) and *Cutibacterium* (OTU_41), were common across different arthropods from different nests (Fig. 5B).

Since our nest material sampling does not fully mirror the arthropod sample set, especially with the lack of nest data from Arizona, we are limited in differentiating the environmental components within all analyzed arthropod microbiomes. In triatomines and ticks, the above-mentioned taxa, commonly found in skin or soil microbiomes (50–53), likely originate from the host-nest ecological interface. Based on the nest material and blood meal analyses, we thus suggest three mutually interrelated bacterial sources for vector microbiomes (Fig. 6): first, the environmental microbiome of their natural habitat, i.e., vertebrate nests and middens, as suggested previously for *Triatoma* (11) and other true bugs (27); second, vertebrate skin and skin-derivate microbiomes (54–56) that vectors encounter while feeding (*Neotoma*, Table S1, data sheet S19); and third, vertebrate pathogens circulating in blood as illustrated here for *Triatoma* on the *Mycobacterium-Dasypus* blood meal link and also supported by the distribution patterns for *Rickettsia* and *Borrelia* OTUs across the unfed ticks (see "Microbiome of the soft tick *Ornithodoros turicata*"). While in *Triatoma* and *O. turicata* microbiomes these bacteria likely represent transitional components, in other

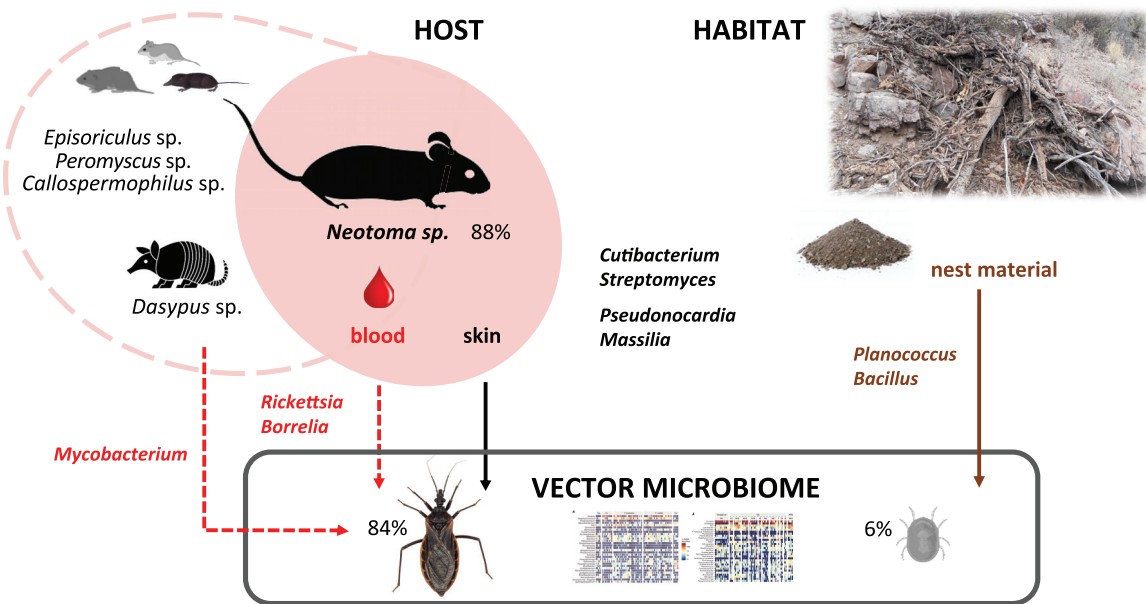

**FIG 6** Scheme for putative sources of vector microbiome based on blood meal analysis and nest microbiome profiling. Percentages next to each vector stand for successfully identified primary blood meal from which 88% matched *Neotoma* sp. The four taxa (*Cutibacterium*, *Streptomyces*, *Pseudonocardia*, and *Massilia*) might originate from both the host skin and the nest microhabitat.

hematophagous vectors, like ticks and lice, vertebrate pathogens gave rise to evolutionarily stable symbiotic associations (57–59).

**Host phylogeny and diet as the microbiome determinants.** Clustering among analyzed arthropod microbiomes based on their similarities (calculated as weighted UniFrac distances, Fig. 7) shows triatomines in a monophyletic cluster and a close relationship between the microbiomes of two Emesinae species. These data suggest compositional convergence of microbiomes in regard to the host phylogenetic distance. The phylogenetic position of *O. turicata* soft ticks as a distant outgroup for the analyzed true bugs is not reflected in the microbiome dissimilarities. *O. turicata* clusters within the hemolymphophagous assassin bugs, resembling the Emesinae microbiomes dominated by common members of the *Rickettsiales*. While we have identified a shared microbiome component between the two hematophagous vectors (see "Blood meal signature among *Triatoma* and tick microbiomes"), it encompasses mostly less-abundant taxa of putatively environmental origin or blood-borne pathogens. These patterns are well reflected by clustering in the nonmetric dimensional scaling (NMDS) analysis and statistical tests supporting the host specificity of *Triatoma* microbiomes. The sample type acts as a significant predictor of microbiome profile, explaining close to 26% of found variation (Table S1, data sheet S24). Diet was identified as another significant predictor, though the effect is comparably lower, accounting for only 6% of found microbiome variation (Table S1, data sheet S24). Both diet and phylogenetic distance as factors underlying Triatominae microbiome structure indirectly support our hypothesis on microbiome change bound to the Triatominae evolutionary shift from hemolymphophagy of predatory Reduviidae to blood feeding.

In summary, comparisons between assassin bugs and triatomines show that the evolution of hematophagy was accompanied by microbiome compositional changes. These changes do not seem to have converged on a preferred taxon set similar to microbiomes in ticks, potentially due to physiological differences in the gut environment of these arthropods and their various feeding behaviors. The observed similarity of microbiomes within the sampled triatomines suggests that though the populations sampled are geographically separated, there is similarity between their communities, indicating convergence of microbial community compositions of this group. Future studies examining the functional diversity of these arthropod microbiomes through metagenomic sequencing and deeper taxon

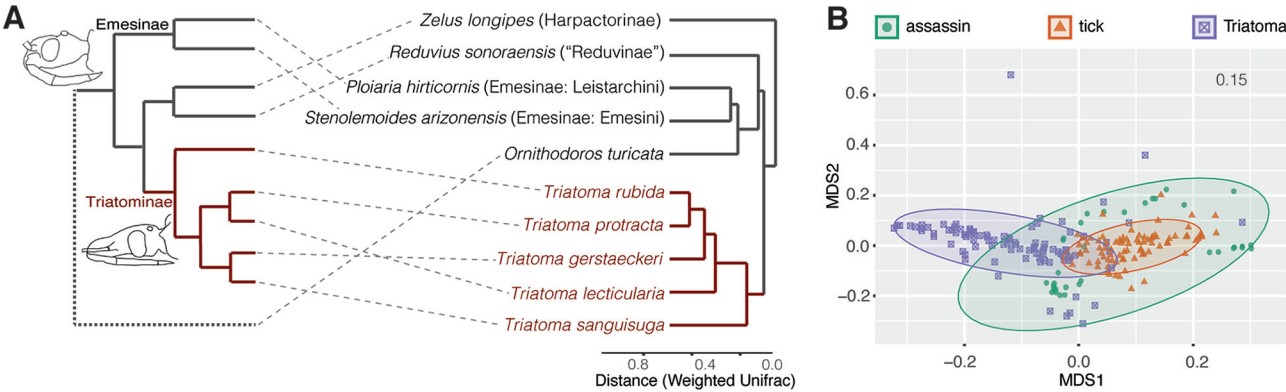

**FIG 7** (A) Schematic host phylogenetic relationships compared to the microbiome dissimilarities based on weighted UniFrac distances. (B) The individual microbiome distances are visualized using NMDS.

sampling will help further clarify how evolution has shaped host-microbiome associations in repeated transitions to hematophagy.

## MATERIALS AND METHODS

**Sample set and study sites.** The study was designed to encompass arthropods collected in the same microhabitat, i.e., white-throated wood rat (*Neotoma albigula*) nests, that share a common phylogenetic (Reduviidae, assassin and kissing bugs) or dietary (blood-feeding vectors, soft ticks and kissing bugs) background. In addition, environmental samples ($n = 24$) of the nest material were collected (Table 1). Two nests (N2 and N41) housed a complete subset of the analyzed arthropod groups (see Table S1, data sheet S1, in the supplemental material). Two hundred thirty-one samples representing different developmental stages of kissing bugs (*Triatoma* sp., $n = 110$), different assassin bug species ($n = 37$), and *Ornithodoros turicata* soft ticks ($n = 84$) were collected through four consecutive years (2017 to 2021) in the states of Arizona ($n = 164$), Florida ($n = 12$), Georgia ($n = 10$), and Texas ($n = 45$) (Table 1). Ten assassin bug individuals were not associated with the nests but collected at the same localities. The localities included Las Cienegas National Conservation Area (LCNCA) and the University of Arizona Desert Station (DS) in Tucson, AZ; Gainesville, FL; Oconee National Forest (GANF) and Sapelo Island, GA; and Chaparral Wildlife Management Area (Chaparral) and Lackland Air Force Base (LacklandAFB), San Antonio, TX. The sites were accessed with permission from the relevant governing bodies (see Acknowledgments). The nest coordinates, sample type, developmental stage, and sex (in adults) were recorded for acquired samples (Table S1, data sheet S1).

**DNA extraction and arthropod species determination.** DNA extraction of samples preserved in absolute ethanol was performed using the DNeasy blood and tissue kit (Qiagen, Hilden, Germany) on whole abdominal tissues according to the manufacturer's instructions, and DNA was stored at −75°C for the downstream molecular analysis. DNA from environmental samples was extracted with the DNeasy PowerSoil Pro kit (Qiagen, Hilden, Germany). Taxonomic determination of sampled ticks was based on their indicative morphological attributes (60). *Triatoma* species were determined based on their morphological and molecular characteristics. In detail, we amplified and Sanger sequenced a 682-bp segment of the *cytB* gene with the previously published (61) primers CYTB7432F and CYTB7433R, which allowed the identification of *T. rubida*, *T. lecticularia*, *T. sanguisuga*, and *T. gerstaeckeri*. Prospective *T. protracta* samples were amplified with primers TprF and TprR (11). Similarly, using previously published primers 18SF and 18SR (62), we obtained partial 18S rRNA gene sequences for assassin bugs. Details on all primers and PCR conditions used are provided in Table S1, data sheet S2. To determine assassin bug taxonomy, 18S rRNA data were aligned with MUSCLE (63) with 14 additional sequences retrieved from NCBI (see Table S1, data sheet S3, for the accession numbers). ModelTest-NG (64) was used to choose the evolutionary model for the phylogenetic inference according to Akaike's information criterion (AIC) (65). The phylogenetic inference was calculated with RAxML (model GTR+I+G4; bootstrap random number seed, 1,234; number of bootstraps, 100; random number seed for the parsimony inferences, 123) (66). For the samples that were morphologically identified as the genus *Reduvius*, we employed an alternative mitochondrial marker (16S rRNA gene amplified with 16sa and 16sb primers [62]) and followed the same approach as described above.

**TABLE 1** Summary of analyzed sample types from different states

| Sample type | No. of samples by state: | | | | |
|---|---|---|---|---|---|
| | Arizona | Florida | Georgia | Texas | Sum |
| *Triatoma* | 85 | 5 | 0 | 20 | 110 |
| Tick | 59 | 0 | 0 | 25 | 84 |
| Assassin bug | 20 | 7 | 10 | 0 | 37 |
| Nest material | 0 | 9 | 0 | 15 | 24 |
| Sum | 164 | 21 | 10 | 60 | 255 |

**Amplicon library preparation.** Amplification of the 16S rRNA V4-V5 region was carried out according to Earth Microbiome Project (EMP) standards (http://earthmicrobiome.org/protocols-and-standards/16s/). Multiplexing of 255 samples and 15 controls utilized a double barcoding strategy including 12-bp Golay barcodes in the forward primer and 5-bp barcodes in the reverse primer. The blocking primer for the 18S rRNA gene was employed as described previously (11). Seven negative controls were amplified along with the samples including two controls for the DNA extraction procedure (Blank2 and Blank21) and five controls from the PCR amplification step (NK, NK1, NK2, NK3, and NK11). Eight positive controls were used to confirm the barcoding output and to assess the detection limit and amplification bias. Positive controls included commercially purchased genomic DNA templates of four mock microbial communities with variable GC content and distribution. ATCC MSA-1000 (samples MCE and MCE1) and ATCC MSA-1001 (samples MCS and MCS1) are composed of 10 bacterial taxa with equal and staggered distributions, respectively. ZymoBIOMICS microbial community DNA standard (samples PC1 and PC3) and ZymoBIOMICS microbial community DNA standard II (samples PC2 and PC31) share eight bacterial and two yeast taxa with even and log distributions, respectively. The amplicons were purified using AMPure XP (Beckman Coulter) magnetic beads and pooled equimolarly. An additional purification step using Pippin Prep (Sage Science) was employed to remove high concentrations of the 18S rRNA gene blocking primer and all unspecific amplification products. The nest material library containing 24 environmental templates was processed according to the same workflow with a single PCR negative control and two positive controls. The libraries were sequenced in two runs of MiSeq (Illumina) using v2 and Nano v2 chemistry with 2- by 250-bp output.

**Analysis of the amplicon data.** Downstream processing, i.e., demultiplexing, merging, trimming, quality filtering, and OTU picking of reads, was performed by implementing corresponding scripts from USEARCH v9.2.64 as previously described (11). Briefly, merged demultiplexed reads from both sequencing runs were joined in a single data set that was subjected to quality and primer trimming resulting in a final amplicon length of 373 bp. The OTU table was created by generating a representative set of sequences based on 100% identity clustering and performing *de novo* OTU picking using USEARCH global alignment at 97% identity match, including chimera removal (67). Taxonomic assignment of representative sequences was executed using the BLASTn algorithm (68) against the sequences of single-subunit (SSU) rRNA genes from the SILVA_138.1_SSUREF_tax database (https://www.arb-silva.de/no_cache/download/archive/release_138.1/Exports/) (69). Filtering of potential contaminants from the OTU table was performed using decontam package v1.18.0 (70) in the R environment (71) based on frequency and prevalence methods of selection (https://rdrr.io/bioc/decontam/man/isContaminant.html), removing 48 out of 4,777 OTUs (Table S1, data sheets S4 and S5). In addition, bacterial taxa of the genus *Sphingomonas* were filtered out as it was a known contaminant in our laboratory (11). A single *Wolbachia* OTU (OTU_95) determined as a contaminant by the decontam package was used for further analysis. The choice to retain *Wolbachia* OTUs in the data set was based on their prevalence among the samples (70/232 samples), including assassin bugs (18/50) and ticks (20/50), for which associations with *Wolbachia* were previously reported (72–74).

Taking advantage of 12S rRNA data present in our data set as a result of nonspecific amplification (75), we identified blood meals from triatomines and ticks. An OTU table was generated from joined and clustered reads by utilizing USEARCH v9.2.64 (67) commands (fastq_join, fastx_uniques, and cluster_otus). Taxonomic assignment of identified OTUs was performed by BLASTn search against the NCBI nucleotide database (76) and restricted to the first hit only. The results were first filtered to include mammals and exclude samples with fewer than 20 reads. In addition, since the hits for the genera *Neotoma* and *Homo* were also observed in some negative controls, blood meal analysis was performed only with the samples where the two taxa reached higher read numbers than those in the negative controls (197 and 45 reads, respectively). The primary blood meal source was determined by the dominant 12S rRNA OTU, while the secondary sources were represented by other highly abundant OTUs found within a particular sample.

**Microbiome analyses.** The profiles of the positive controls were compared and plotted using the ggplot2 v3.4.0 (77) and svglite v2.1.0 (78) packages. Additional cleanup, microbiome analyses, data visualization, and statistical tests were performed in the R environment (71) using phyloseq v1.42.0 (79), vegan v2.6-4 (80), and MicEco v0.9.19 (https://github.com/Russel88/MicEco) and MicroEco v0.13.0 (81) packages. Graphical outputs were generated utilizing ggplot2 v3.4.0 (77) and further processed in Adobe Illustrator v25.4.1. A "clean" data set was prepared by filtering out any archaeal, eukaryotic, mitochondrial, and chloroplast OTUs. Since the environmental samples produced considerably smaller amounts of data, we have analyzed the "clean" data set at two rarefaction levels (800 and 1,000 reads per sample, seed = 5) to retain enough of the nest material samples. We focused on effects of several microbiome determinants including environmental microbiome background of sampled habitats, host phylogenetic origin, and the dietary niche (blood versus hemolymph and different blood meal sources).

The overall consistency of Triatominae microbiome profiles sequenced here with previously published data (11) was confirmed based on their taxonomic composition and dissimilarities found among *Triatoma* species. In the MicroEco v0.13.0 (81) package, we used the trans_abundance class to determine highly abundant bacteria. Alpha diversity was calculated by utilizing the trans_alpha class, and differences found among species were statistically evaluated based on Dunn's Kruskal-Wallis multiple-comparison method. Using the trans_beta class, beta diversity analysis was performed based on Bray-Curtis distances among individual microbiomes and visualized through nonmetric dimensional scaling (NMDS) ordination. Adonis2 implemented under the cal_manova method of the trans_beta class was employed to confirm *Triatoma* species as a statistically significant factor driving the microbiome profile. Similarly, using trans_abundance, trans_alpha, and trans_beta classes, we have calculated alpha and beta diversity for tick, assassin bug, and nest material microbiomes, including statistical evaluations for selected factors potentially underlying found variation, and we produced heatmaps to visualize their taxonomic composition.

The ps_venn function in the MicEco package was utilized first to identify the potentially environmentally acquired fraction of analyzed invertebrate microbiomes (defined as OTUs present in at least 30% of all nest material samples [Table S1, data sheet S6]). Second, unique and shared taxa within and among different sample groups were identified under a more stringent fraction range (0.5, 0.8, 0.9, and 1 for all samples within the group). For the *Triatoma* shared microbiome, the ps_venn results were visualized using the UpSet plot (https://github.com/visdesignlab/upset2) (82). The ps_venn function was further employed for identifying possible taxonomic convergence among microbiomes of blood-feeding vectors (*O. turicata* and *Triatoma* spp.), related Reduviidae (kissing and assassin bugs), and within the two Reduviidae groups, i.e., among different *Triatoma* species, among different assassin species (*Reduvius sonoraensis* ["Reduvinae"], *Zelus longipes* [Harpactorinae], *Ploiaria hirticornis* [Emesinae: Leistarchini], and *Stenolemoides arizonensis* [Emesinae: Emesini]).

**Phylogenetic inference of putative symbiotic taxa shared by different hosts.** The analysis comprised OTUs shared between at least two different sample types (triatomines, ticks, and assassins) that belong to the bacterial genera previously reported as arthropod symbionts. The data sets were generated for the families *Midichloriaceae*, *Morganellaceae*, and *Rickettsiaceae* and for the genus *Asaia* from the OTU representative sequences and partial 16S rRNA gene sequences downloaded from NCBI. The data sets were aligned with MUSCLE software (63). For each alignment, an evolutionary model was selected, and phylogenetic analysis was performed as described above for the arthropod species determination.

**Data availability.** Raw sequence reads generated from this study were deposited in the NCBI Sequence Read Archive (SRA) repository under BioProject accession no. PRJNA898622. The complete R code employed in this study with its associated data sets is available at https://github.com/hassantarabai/Convergency-MS-2022.

## SUPPLEMENTAL MATERIAL

Supplemental material is available online only.
**SUPPLEMENTAL FILE 1**, PDF file, 1.8 MB.
**SUPPLEMENTAL FILE 2**, XLSX file, 4.2 MB.

## ACKNOWLEDGMENTS

We acknowledge the University of Arizona Department of Entomology's Desert Station for graciously supporting our field activities. We thank Samantha M. Wisely and Chanakya R. Bhosale for logistical and technical assistance with sampling in Gainesville, FL. We further acknowledge the contributions of the Texas Park and Wildlife Management Department and the Bureau of Land Management (Gila District Office, Tucson, AZ).

This work was supported by the Czech Science Foundation (grant number 21-10185M to E.N.).

E.N. conceived the study. E.N., A.M.F., J.J.B., N.F., R.L.S., N.L.B., K.J.V., and W.R. participated in planning and executing field collections. J.Z. and N.F. performed the DNA isolation, library preparation, and sequencing. H.T., A.M.F., and E.N. analyzed the data and interpreted results. H.T., A.M.F., and E.N. wrote the draft, and all the authors contributed to its improvement. All the authors read and approved the final manuscript.

We declare that we have no competing interests. Robert L. Smith has not sought nor has he received any remuneration for providing access to Desert Station and adjacent private land and residence, nor has he received payments from any researchers or scholarly visitors on Desert Station.

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
