## [Reviewer comments · Microbiology Spectrum]

Microbiology Spectrum

Microbiomes of blood feeding triatomines in the context of their predatory relatives and the environment

Hassan Tarabai, Anna Floriano, Jan Zima Jr., Natalia Filova, Joel Brown, Walter Roachell, Robert Smith, Norman Beatty, Kevin Vogel, and Eva Novakova

Corresponding Author(s): Eva Novakova, Jihoceska Univerzita v Ceskych Budejovicich Prirodovedecka Fakulta

Review Timeline:

Submission Date:	April 21, 2023
Editorial Decision:	May 11, 2023
Revision Received:	May 18, 2023
Accepted:	May 23, 2023

Editor: Denis Sereno

Reviewer(s): The reviewers have opted to remain anonymous.

Transaction Report:

DOI: <https://doi.org/10.1128/spectrum.01681-23>

May 11, 2023

Prof. Eva Novakova
Jihoceska Univerzita v Ceskych Budejovicich Prirodovedecka Fakulta
Branisovska 1760
Ceske Budejovice 37005
Czech Republic

Re: Spectrum01681-23 (Microbiomes of blood feeding triatomines in the context of their predatory relatives and the environment)

Dear Prof. Eva Novakova:

Thank you for submitting your manuscript to Microbiology Spectrum. When submitting the revised version of your paper, please provide (1) point-by-point responses to the issues raised by the reviewers as file type "Response to Reviewers," not in your cover letter, and (2) a PDF file that indicates the changes from the original submission (by highlighting or underlining the changes) as file type "Marked Up Manuscript - For Review Only". Particularly a "data statement availability is lacking" please provide it along with links to reach them. Please use this link to submit your revised manuscript - we strongly recommend that you submit your paper within the next 60 days or reach out to me. Detailed instructions on submitting your revised paper are below.

Link Not Available

Sincerely,

Denis Sereno

Journals Department
Reviewer comments:

Reviewer #1 (Public repository details (Required)):

The study produced a significant amount of sequencing data, that should be submitted to GenBank.

Reviewer #1 (Comments for the Author):

Overall, the study is original in its design and has produced interesting results. I do not see any significant biases warranting a revision or including additional data. The results are clear and valuable for any future paratransgenesis efforts and an excellent

contribution to the knowledge of the microbiome of five triatomine species in the southern US.

The few weaknesses of the study were the failed attempt to determine the potential sources of the OTU's was not attained since the phylogenetic resolution obtained from the data did not produce fully resolved trees (supplementary figures 5-7). However, this is probably due to insufficient phylogenetic informative sites at the molecular marker used for the study.

I only found a couple of editorial errors, which are the following.

Line 69: Change "dietary constraint" to "constraints"

Line 73: There is an extra space after the cited references.

Line 320: Change "corelate" with "Correlate"

Reviewer #2 (Comments for the Author):

Please see the attached file.

Staff Comments:

Preparing Revision Guidelines

Please return the manuscript within 60 days; if you cannot complete the modification within this time period, please contact me. If you do not wish to modify the manuscript and prefer to submit it to another journal, please notify me of your decision immediately so that the manuscript may be formally withdrawn from consideration by Microbiology Spectrum.

Revision Spectrum 01681-23

Microbiomes of blood feeding triatomines in the context of their predatory relative and the environment

In this manuscript, the authors investigate microbiome signatures of five *Triatoma* species vector of *Trypanosoma cruzi* (Chagas disease). It is an interesting and well-written study that deserves to be published. I have only a few minor comments for the authors.

Methods

- Lines 113-120: consider putting this information in a summary table and including it in the manuscript.
- Line 130: absolute or 70% ethanol?
- Lines 136-144: consider adding more information about the conditions for PCR and sequencing reactions.
- Line 156: 255 or 265 samples?

231 kissing bugs
- 110 Triatoma
- 37 assassin bugs
- 84 Ornithodoros turicata soft ticks
24 additional environmental samples
10 assassin bugs no associated with vertebrate nests

Results and discussion

- At some points of this section, I got lost in the objective of the study.
- I am not sure if these two sections can be presented together.
- Consider adding a concluding paragraph/section.
- Line 351: Figure 3C appears before 3A and 3B.

Response to Reviewers

Spectrum01681-23 (Microbiomes of blood feeding triatomines in the context of their predatory relatives and the environment)

We thank both the reviewers for valuable comments and suggestions that have facilitated improvements to this manuscript. Below, we provide point-by-point responses and describe all the modifications made throughout the manuscript (highlighted in the "Marked Up Manuscript - For Review Only" pdf). Our responses are printed in **bold italics**, quotations from the manuscript are in *italics* (where line numbers are given, they correspond to the revised version).

Reviewer comments:

Reviewer #1 (Public repository details (Required)):

The study produced a significant amount of sequencing data, that should be submitted to GenBank.

Possibly, the corresponding author missed a dedicated statement during the submission process. The submitted manuscript contained Data Availability section on lines 548-552 quoted below. The functionality of the BioProject Number as well as the github link have been reverified. The statement now appears on lines 547-551.

"Data Availability"

"Raw sequence reads generated from this study were deposited in the NCBI Sequence Read Archive (SRA) repository under BioProject PRJNA898622. The complete R code employed in this study with its associated datasets are available at <https://github.com/hassantarabai/Convergency-MS-2022>."

Reviewer #1 (Comments for the Author):

Overall, the study is original in its design and has produced interesting results. I do not see any significant biases warranting a revision or including additional data. The results are clear and valuable for any future paratransgenesis efforts and an excellent contribution to the knowledge of the microbiome of five triatomine species in the southern US.

The few weaknesses of the study were the failed attempt to determine the potential sources of the OTU's was not attained since the phylogenetic resolution obtained from the data did not produce fully resolved trees (supplementary figures 5-7). However, this is probably due to insufficient phylogenetic informative sites at the molecular marker used for the study.

We agree that the poorly resolved OTU based phylogenies represent a slight limitation. We agree that that this is due to an insufficient phylogenetic signal within 16S rRNA gene amplicons as originally acknowledged (365-367).

I only found a couple of editorial errors, which are the following.

Line 69: Change "dietary constraint" to "constraints"

Line 73: There is an extra space after the cited references.

Line 320: Change "corelate" with "Correlate"

We thank the reviewer for pointing these out. We have corrected these, double spaces, and other typos as follows (corrections are underscored):

Line 51: Ornithodoros

Line 68: constraints

Line 73: leeches

Line 121: Wildlife

Line 160: variable

Line 262: signals

Line 312: Similar

Line 443: OTU 41

Reviewer #2 (Comments for the Author):

In this manuscript, the authors investigate microbiome signatures of five *Triatoma* species vector of *Trypanosoma cruzi* (Chagas disease). It is an interesting and well-written study that deserves to be published. I have only a few minor comments for the authors.

Methods

- Lines 113-120: consider putting this information in a summary table and including it in the manuscript.

We do agree that there is a considerable number of different samples. To ease the readers insight, we have added a summary table (Table 1, line 493).

- Line 130: absolute or 70% ethanol?

We have clarified the information (line 128).

- Lines 136-144: consider adding more information about the conditions for PCR and sequencing reactions.

We have added PCR conditions for each set of primers in the Supplementary Table 1-DATASHEET S2. Line 138 and 139 now reads: "Details on all used primers and PCR conditions are provided in Supplementary Table 1-DATASHEET S2."

- Line 156: 255 or 265 samples?

231 kissing bugs
- 110 Triatoma
- 37 assassin bugs
- 84 Ornithodoros turicata soft ticks
24 additional environmental samples
10 assassin bugs no associated with vertebrate nests

We agree that the original text was misleading. We have rephrased the text (lines 114-119) and provided a summary table as suggested above.

Results and discussion

- At some points of this section, I got lost in the objective of the study.

We have made several adjustments throughout the text to make it more concise (all visible in the “Marked Up Manuscript - For Review Only”) and most importantly added a concluding paragraph that we believe contributed to the clarity of our objective (lines 481-490).

- I am not sure if these two sections can be presented together.

Since the manuscript is quite complex, comprising analyses of different sample types, we have decided to use the combined format of Results and Discussion. We believe, ASM Journals have been following the format neutral policy. We also provide examples of similarly formatted papers that have been published in Microbiology Spectrum, e.g. [10.1128/spectrum.04478-22](https://doi.org/10.1128/spectrum.04478-22), [10.1128/spectrum.00247-22](https://doi.org/10.1128/spectrum.00247-22), [10.1128/spectrum.01915-21](https://doi.org/10.1128/spectrum.01915-21), [10.1128/spectrum.04353-22](https://doi.org/10.1128/spectrum.04353-22).

- Consider adding a concluding paragraph/section.

We have added a concluding paragraph (lines 481-490).

- Line 351: Figure 3C appears before 3A and 3B.

The section Microbiome of the soft tick Ornithodoros turicata (lines 340-360) has been restructured to ease the readers understanding and to comply with the panel layout of Figure 3.

May 23, 2023

Prof. Eva Novakova
Jihoceska Univerzita v Ceskych Budejovicich Prirodovedecka Fakulta
Branisovska 1760
Ceske Budejovice 37005
Czech Republic

Re: Spectrum01681-23R1 (Microbiomes of blood feeding triatomines in the context of their predatory relatives and the environment)

Dear Prof. Eva Novakova:

Your manuscript has been accepted, and I am forwarding it to the ASM Journals Department for publication. You will be notified when your proofs are ready to be viewed.

Sincerely,

Denis Sereno
Editor, Microbiology Spectrum
